# Urban land patterns can moderate population exposures to climate extremes over the 21st century

Jing Gao [1,3] ✉ & Melissa S. Bukovsky [2,3] ✉

Climate change and global urbanization have often been anticipated to increase future population exposure (frequency and intensity) to extreme weather over the coming decades. Here we examine how changes in urban land extent, population, and climate will respectively and collectively affect spatial patterns of future population exposures to climate extremes (including hot days, cold days, heavy rainfalls, and severe thunderstorm environments) across the continental U.S. at the end of the 21st century. Different from common impressions, we find that urban land patterns can sometimes reduce rather than increase population exposures to climate extremes, even heat extremes, and that spatial patterns instead of total quantities of urban land are more influential to population exposures. Our findings lead to preliminary suggestions for embedding long-term climate resilience in urban and regional land-use system designs, and strongly motivate searches for optimal spatial urban land patterns that can robustly moderate population exposures to climate extremes throughout the 21st century.

Urban areas across the world are increasingly affected by climate extremes[1]. Over the coming decades, climate change is projected to increase the frequency and the intensity of extreme weather[2], and global urbanization is expected to bring about further population concentration and urban land expansion[3,4]. These mechanisms interact, and as a result, urbanization has usually been reported to escalate population exposures (frequency and intensity) to climate extremes[5–7] (e.g., when urban lands enhance climate extremes at places where population concentrates); however, urban land patterns are human designs that can be changed. During the 21st century, many new urban centers will be built and existing ones renovated. In order to take these opportunities to embed long-term climate resilience in urban land patterns, we must first understand whether it is possible for urban land patterns to moderate, rather than amplify, population exposures to climate extremes. Here we investigate how climate change and urbanization, respectively and collectively, affect spatial population exposure to four climate extremes – hot days, cold days, heavy rainfalls, and severe

thunderstorm environments (STEs) – across the continental U.S. at the end of the 21st century.

Compared to existing literature, this research emphasizes three aspects of future population exposures to climate extremes: (i) Effects of urban land change. Change in urban land has usually been overlooked in regional climate projections[8]. But this practice should be challenged for both developed and developing countries[9]. By 2100, the U.S. total amount of urban land could be 4.6 times the amount in 2000 under a high urban expansion scenario and 2.6 times under a business as usual scenario[4]. This level of urban land expansion can alter regional-scale climate conditions[8]. (ii) Continental-scale spatial patterns. Though urban lands' modifying effects on climate extremes are recognized[10–12], much of the existing literature has focused on individual cities[6,13,14]. Here we examine population exposures to climate extremes across the continental U.S., by different urban development densities, climate regions, and 109 sizeable urban centers, to understand how the regional-scale climate signals from anticipated urban land change may be experienced by future population. (iii) Population

[1]Department of Geography and Spatial Sciences & Data Science Institute, University of Delaware, Newark, DE 19716, USA. [2]Haub School of Environment and Natural Resources, University of Wyoming, Laramie, WY 82072, USA. [3]These authors contributed equally: Jing Gao, Melissa S. Bukovsky.
✉e-mail: jinggao@udel.edu; melissa.bukovsky@uwyo.edu

exposures to climate extremes beyond temperature. Existing literature on future population's vulnerability to climate extremes has primarily focused on temperature[7,15,16]. However, patterns of extreme precipitation and convective environments will also evolve, as both climate change and urbanization modify future land-atmosphere dynamics[17–22]. Effects of urban heat island (UHI) not only strengthen sensible heat fluxes to increase temperature, but also enhance surface roughness and atmospheric circulations to influence the frequency and the intensity of severe convective storms[23], including extreme precipitation and thunderstorms. Here we examine hot days, cold days, heavy rainfalls, and STEs.

We analyze three future scenarios combining different climate, land use, and population conditions at the end of the 21st century (EOC; 2075–2100 summaries) for spatial population exposures to the four climate extremes at a 25-km resolution (more information in Methods and SI Table 1): (i) high greenhouse gas induced warming with no population nor urban land change (denoted as "climate", showing effects from climate change only), (ii) high greenhouse gas induced warming and high population growth with no urban land change ("climate+pop", the current common practice), and (iii) high greenhouse gas induced warming, high population growth, and high urban land expansion ("climate+pop+landUse", enabling investigation of urban land change effects). We recognize that high scenarios (details in Methods) may not be the most likely scenarios. They are used here, because this research focuses on climate extremes, and high scenarios provide more data points on extreme cases, to support later trend identification and pattern comparison of influences from climatic and social systems and their interactions. Population exposures at the beginning of the 21st century (BOC; 1980–2005 summaries) (denoted as "historical") were also analyzed as comparison baselines.

## Results

### Exposure by urban development densities

In our scenarios, over the 21st century, the U.S. average temperature increases by 4.4 °C, the national population becomes 2.3 times the BOC level, and the total amount of urban land 4.2 times. Together they make the EOC population exposure to hot days 13.6 times the BOC level, cold days 1.1 times, heavy rainfalls 7.5 times, and STEs 2.5 times.

We find that climate trends determine the spatial distributions of extreme weather, population trends determine the overall magnitudes of exposures (e.g., places with no residents will have no population exposure regardless of whether climate extremes happen), and urban land trends affect how people experience climate extremes (details below).

We capture effects of changes in climate, urban land, and population over the 21st century as ratios between exposure counts (ECs) from different scenarios, because such ratios can robustly separate effects of climate, urban land, and population changes (more explanations in Methods). We measure effects of climate change from BOC to EOC on population exposure to a climate extreme as $(EC_{climate}/EC_{historical})$, effects of urban land expansion as $(EC_{climate+pop+landUse}/EC_{climate+pop})$, and effects of population change as $(EC_{climate+pop}/EC_{climate})$.

According to these ratios (Fig. 1, SI Table 2), urban land effects on future population exposures to climate extremes are generally small at the national total level, though somewhat higher for temperature extremes (hot and cold days). That is, urban land expansion alone has limited impact on the national total count of population exposure.

However, at the individual level, continued urbanization strongly affects future generations' experiences of climate extremes. As urban land keeps expanding, exposures in non-urban areas decrease, while exposures in urban areas, especially those with mid-to-high development densities, increase (Fig. 1). For all four climate extremes, the dominant majority of BOC population exposures are non-urban (roughly 65–75%), while EOC exposures are primarily urban (roughly 50–75%) (SI Table 2). That is, more and more people will experience climate extremes as urban residents.

Urbanization can increase an average individual's exposure to climate extremes. For example (SI Table 2), the BOC U.S. national per capita exposure (i.e., total exposure divided by population) to hot days is 7 days per person per year, while the EOC exposure is 32 days per person per year when considering only climate change, and 41 days per person per year when considering both climate change and urbanization. The EOC per capita exposure also differs substantially across different urban development densities: in non-urban areas, it is 29 days per person per year, and in mid-to-high-density developments, 53 days per person per year. These results illustrate the necessity of

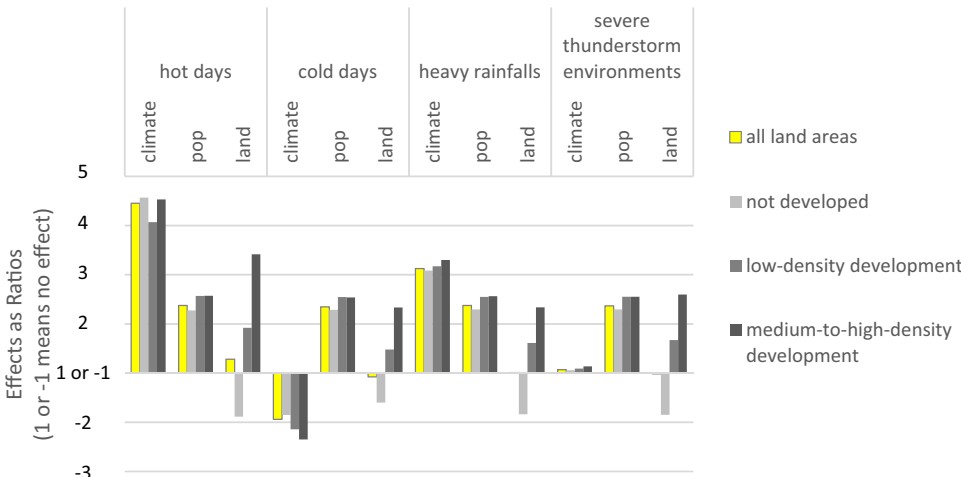

**Fig. 1 | Effects of 21st-century climate, population, and urban land changes on population exposures to four climate extremes in areas with different urban land development densities.** Effects from the beginning (BOC) to the end (EOC) of the 21st century are measured as ratios: When an effect increases exposure, it is reported as $\left(\frac{exposure_{EOC}}{exposure_{BOC}}\right)$; when an effect decreases exposure, it is reported as $\left(-\frac{exposure_{BOC}}{exposure_{EOC}}\right)$. For example, the climate effects for hot days across all land development densities is 4.5, meaning that, over the 21st century, climate change alone increases the U.S. total population exposure to hot days to 4.5 times the BOC total exposure; in contrast, the climate effects for cold days across areas with low development densities is −2.1, meaning that, over the 21st century, climate change alone reduces population exposure to cold days in areas with low-density development by $\frac{1}{2.1}$, i.e., about half. A ratio of 1 or −1 means there is no change in exposure counts, i.e. no effect.

considering urban land change when investigating future population exposures to climate extremes.

Surprisingly, urban land expansion can also decrease average exposures to climate extremes in some cases. For example (SI Table 2), urban land expansion in mid-to-high-density developments decreases the EOC per capita exposure to heavy rainfalls. This change in exposure might be caused by physical land-atmosphere mechanisms (e.g., UHI effects can strengthen convective inhibition over densely developed urban areas[8], which can decrease the frequency of extreme precipitation events to counteract the broadly-occurring enhancing effect of climate change), and/or spatial population-urban-land dynamics (e.g., many densely developed urban areas are not residential and hence show low population exposures). Identifying what patterns and mechanisms can reduce population exposures to climate extremes, and using that knowledge to inform urban designs, are our planned next research steps. The significance of this work is that the results show carefully designed urban land patterns are able to practically moderate instead of intensify the nation's future experiences of climate extremes.

Meanwhile, climate effects are more prominent than urban land effects to influence total population exposure counts (Fig. 1). Climate effects are highest for hot days, then heavy rainfalls, lower for cold days, and lowest for STEs. For all four weather extremes, climate effects show a subtly increasing trend from low to high urban development densities.

On a different note, population effects, regardless of extreme weather types, urban development densities, climate regions, and individual urban centers, are always around the national total population change ratio over the 21st century (i.e., 2.3) (e.g., Fig. 1), suggesting that, the EOC population maps we used are scaled versions of the BOC population condition, and the spatial distribution of relatively high and low values barely changes over time. Such stationarity results from underlying assumptions of commonly-used spatial population models for long-term projections (i.e., gravity models). Though a known limitation, the practice has been assumed

innocuous. However, our results here show the assumption is so influential that it dominates any spatial patterns projected by gravity models, and therefore, current spatial population projections are insufficient for analyzing possible changes in spatial patterns. Below, we focus our discussion of spatial variations on climate and urban land effects, while recognizing population determines the overall magnitude of exposure.

## Exposure by climate regions

Comparing population exposures in different climate regions at the beginning versus the end of the 21st century (Fig. 2), major changes in regional patterns are present for hot days and heavy rainfalls, and the same regional patterns roughly remain for STEs and cold days.

For hot days, the ranking of regions by their exposure counts shuffle from BOC to EOC: Some regions that do not rank highly at BOC are anticipated to become new hotspots of heat exposure at EOC (Fig. 2, SI Fig. 4), e.g., the Southeast is expected to substantially surpass the Southwest to become the most heat-stressed region in the country, the Southern Great Plains catch up to the Southwest, while the Northeast and the Midwest also experience considerable increases in heat exposure. The drastic changes in the ranking of regions are important, since planning and distributing resources according to past priorities might lead to additional vulnerabilities for the future, especially for the regions rising in priorities.

For heavy rainfalls, drastic increases in exposure are expected for most regions except the great plains. This change in regional patterns is a result of changes in both society and climate, as regions with similar climate effects do not necessarily show similar changes in exposure (e.g., the Northwest versus the Southwest in Fig. 2).

Comparing climate effects with urban land effects in different climate regions, we find that neither the magnitude nor the sign of urban land effects are clearly associated with those of climate effects, suggesting that urban land effects are not derivatives of climate effects, and the possibility of using urban land to moderate population

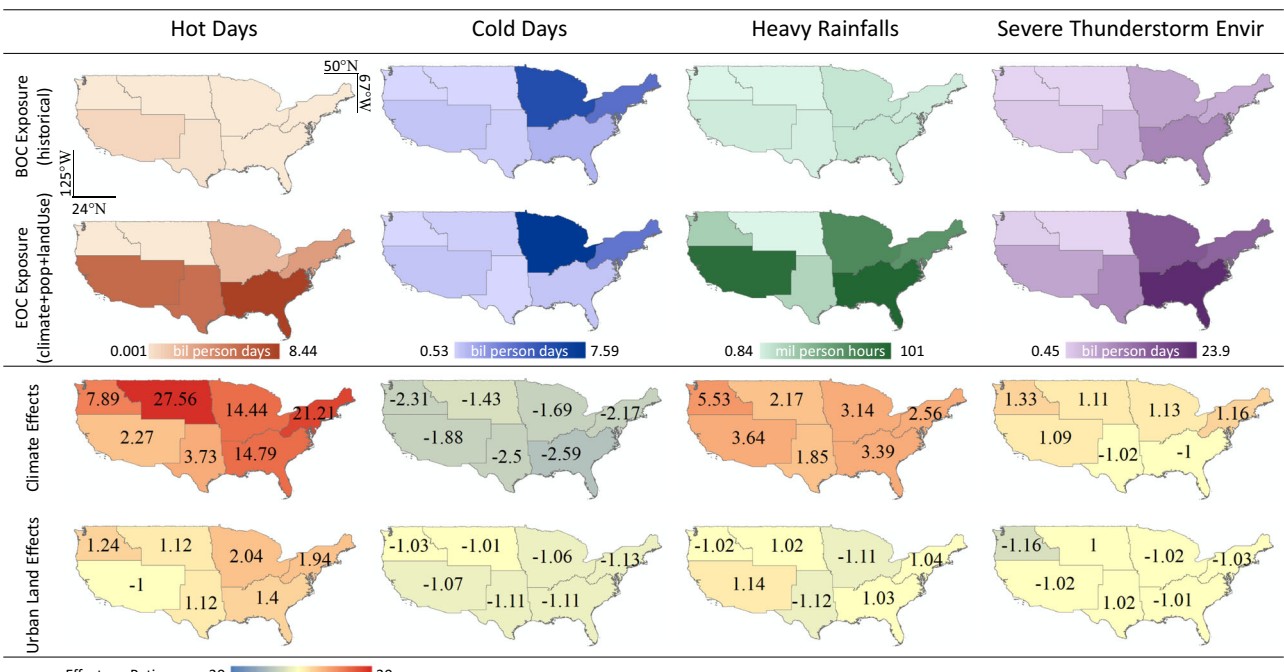

**Fig. 2 | Population exposures to four climate extremes in different climate regions at the beginning (BOC) and the end (EOC) of the 21st century, and effects of 21st-century climate change versus urban land expansion on such exposures.** Effects are measured as ratios in the same way as for Fig. 1: When an

effect increases exposure, it is reported as $\left(\frac{exposure_{EOC}}{exposure_{BOC}}\right)$; when an effect decreases exposure, it is reported as $\left(-\frac{exposure_{BOC}}{exposure_{EOC}}\right)$. The continental U.S. spans approximately 25–49°N latitude and 67–125°W longitude.

exposures to climate extremes likely exists for all climate regions. For example, for hot days, regions with the highest climate effects do not coincide with the highest urban land effects; for heavy rainfalls, the Southwest and the Midwest experience similar climate effect ratios but their urban land effects show opposite signs (Fig. 2).

Comparing across regions, the Southeast is becoming the most exposed to three of the four climate extremes; the Northeast ranks high (but not the highest) for all four extremes; the Midwest experience is similar to the Northeast with the most exposure to cold days; the Southwest will be primarily stressed by hot days and heavy rainfalls; the Southern Great Plains' main challenge will be heat (like now); the Northwest and the Northern Great Plains are anticipated to have

the least population exposures to climate extremes due to both climatic and urban processes.

## Exposure in sizable urban centers

According to the United Nations' World Urbanization Prospects[24], 143 sizeable cities (i.e., with a population size greater than 300,000) exist in the continental U.S. Some adjoin each other, and after consolidating their merging boundaries, 109 urban centers remain (SI Fig. 3) (details in "Methods").

Examining the 109 urban centers (Fig. 3), we find urban land effects can decrease as well as increase population exposures to climate extremes, and urban land effects can decrease population

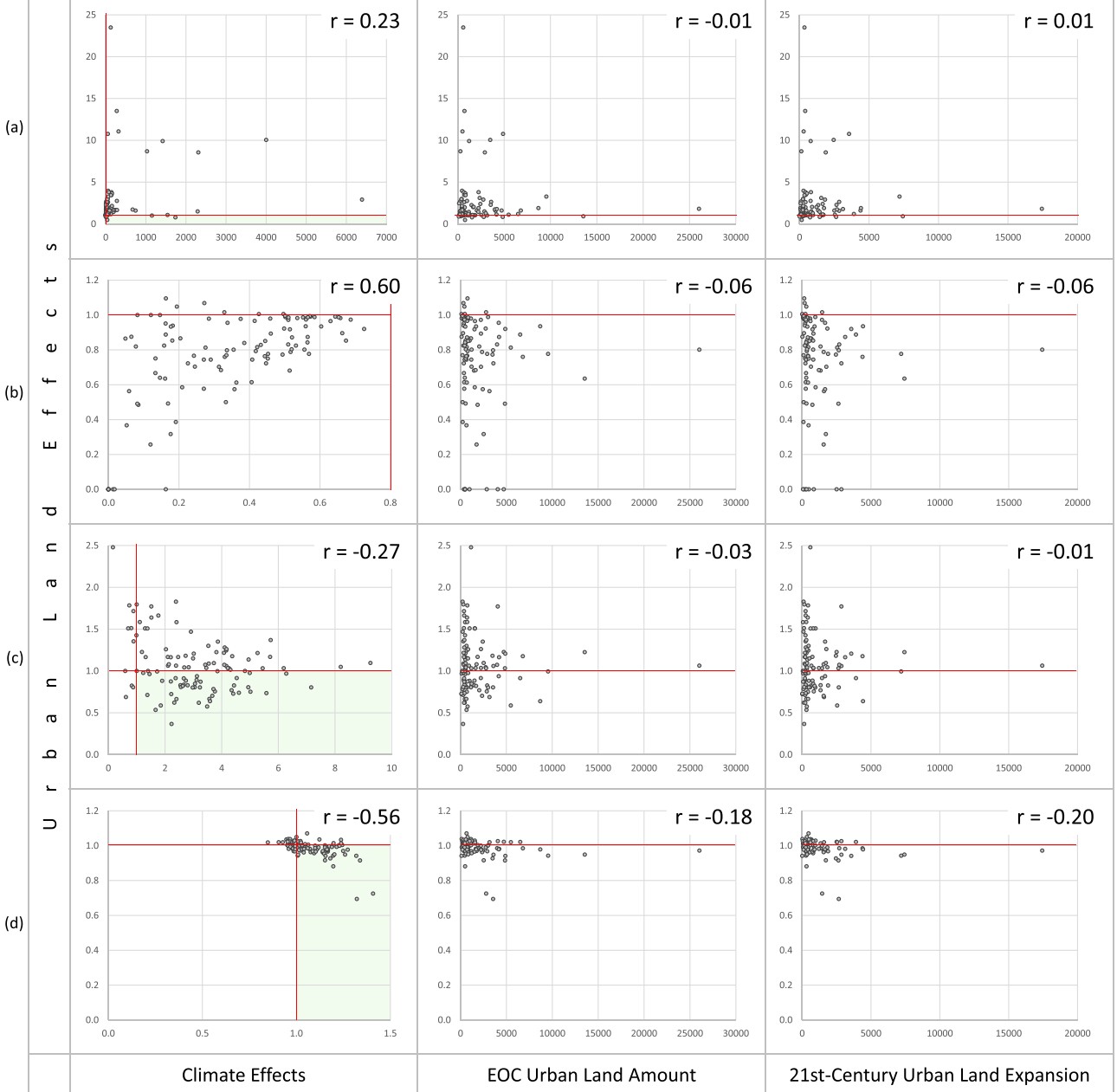

**Fig. 3 | Scatterplots and correlation coefficients pairing urban land effects with climate effects, urban land total amounts at the end of the 21st century, and urban land change amounts from the beginning (BOC) to the end (EOC) of the 21st century, across 109 sizeable urban centers, for four climate extremes. a** hot days, **b** cold days, **c** heavy rainfalls, and **d** severe thunderstorm environments. Effects are measured as ratios $\left(\frac{exposure_{EOC}}{exposure_{BOC}}\right)$. When a ratio > 1, the effect increases exposure; when a ratio < 1, the effect decreases exposure. Each point represents

one sizeable urban center. Points that fall below or to the left of the red lines (where the ratios = 1) reflect effects that decrease exposure (either from urban land expansion or climate change). Points that fall in the light green shaded areas are urban centers where climate change increases exposure and urban land expansion decreases exposure, that is, where urban land patterns moderate population exposures to climate extremes.

exposures in urban centers where climate effects increase exposures. That is, it is possible for urban land effects to moderate climate effects on population exposures to climate extremes at the city level.

Moreover, urban land effects show no correlation with urban land total amounts nor change amounts (Fig. 3), suggesting neither is a key determinant of urban land effects at the city level. Then, only two factors remain to possibly be influential:

First, climate. Not only some low-level correlation exists between climate effects and urban land effects (Fig. 3), but also as climate changes, effects of the same urban land pattern change. In Fig. 3, urban centers with little urban land change exhibit a wide range of urban land effects – since these urban centers experience little change in urban land, the varying urban land effects can only be a result of changing climate-urban-land interactions due to climate change.

Second, urban land patterns (including both the spatial arrangements of urban lands and the types of land covers that urban expansions take over). Because urban land quantities have been ruled out, urban land patterns are the only aspect of urban expansion that can be a key determinant. Identifying what urban land patterns can maximally moderate a city's experience of its evolving climate extremes is an important next step for future research.

Comparing the 109 urban centers (Fig. 4), urban land effects are generally smaller than climate effects, while the two effects are not strongly associated (in terms of both signs and magnitudes). For temperature extremes, urban land expansion generally leads to warming, due to UHI effects, but exceptions exist for a few urban centers, to reduce exposure to hot days and increase exposure to cold days. Though these exceptions are of small magnitudes, their existence showcases the possibility of using urban land effects to moderate, or at least not horribly amplify, climate effects. Especially for hot days, a total of seven urban centers' land changes reduce exposures, i.e., Savannah, GA, Los Angeles, CA, Boise, ID, Seattle, WA, San Francisco, CA, Salt Lake City, UT, and Virginia Beach, VA, with their urban land effects on population exposures to hot days ranging from 0.92 to 0.48. This is NOT to say that the current urban land patterns of these cities already moderate population exposures, but rather their urban land changes simulated in our hypothesized high urbanization scenario show that effect. Investigations of what mechanisms drive the exposure reductions are planned for our next research. We will consider both physical factors (e.g., changes in evapotranspiration, albedo, sea breeze) and population-land dynamics (e.g., infill development, migration). For heavy rainfalls, urban expansions show mixed effects, increasing and decreasing exposures in different urban centers, regardless of climate regions. For STEs, no strong patterns of increase or decrease are present, considering all effects are small.

These results indicate a loose association between urban land effects and climate trends, suggesting the potential of urban land patterns to moderate population exposures to climate extremes at the city level is not dependent on climate effects and is non-trivial. Especially, the urban land projections used in this research are produced by a novel data-driven model[4,25,26] (more information in Methods): While data-driven models are good at capturing various underlying processes and their transitions from data, in this research the projected future urban land patterns would not have reflected conscious land-use designs for long-term climate resilience, because such designs have rarely happened in recent past decades and hence have little reflection in existing data. In the future, as we more intentionally design our urban and regional land-use patterns, their desirable moderating effects on population exposures to climate extremes will likely be larger than what have been observed in this research.

## Discussion

Urban land patterns can moderate instead of amplify population exposures to climate extremes, and spatial patterns rather than total quantities of urban land are more influential for building long-term climate resilience in urban designs. These findings run counter to common beliefs, because existing literature has emphasized almost exclusively the amplifying effects from the increasing amount of urban lands on climate extremes[1,2]. Recognizing that urban land patterns can moderate population exposures to climate extremes encourages researchers and practitioners from a wide range of related fields to design and implement urban and regional land systems with embedded long-term climate resilience, instead of focusing solely on the quantity of urban lands, which may be limited for generating desired sustainability outcome and goes against the natural trend of 21st-century socioeconomic processes.

For the coming decades, continued urbanization is anticipated across the world[3,4], while increasingly severe climate disasters might animate permanent population migration (in addition to temporary displacement) to form new urban centers[27], and motivate renovations of existing urban areas to adapt to the 21st-century climate among other environmental and socioeconomic changes[28]. These are opportunities to build long-term climate resilience into our future urban and regional land systems.

We should hold realistic expectations: Urban land designs cannot eradicate climate extremes, since urban land effects are usually smaller in magnitude than climate effects. However, urban land patterns can substantially alter how most individuals experience future climate extremes. It is therefore worthwhile and beneficial to identify urban land patterns and related physical and socioeconomic mechanisms that can reduce population exposures to climate extremes over the long term, and use such understandings to guide urban designs for the future.

A keyword here is "long-term". Much of current urban systems' climate challenges root from the fact that many contemporary urban developments were designed for the 20th-century climate with no or only short-term climate consideration. However, urban developments, once built, usually stand for decades to hundreds of years. To achieve long-term resilience and prosperity, careful planning for equally long term is necessary. Rather than assuming a fixed pathway into the far future, we should identify policies and decisions that are robust across a wide range of possible future scenarios, invest in urban designs that allow effective adjustments along the way for many possible climate futures, and avoid those that rule out solutions potentially needed down the road.

To identify and implement adaptive urban design pathways, we need a sandbox for iteratively (re-)evaluating alternative design options using agile computation, where long-term futures of large-scale interactions between climate change and urban land patterns can be explored. Especially, we need long-term, large-scale spatiotemporal modeling of influential socioeconomic processes (e.g., urbanization). Due to the challenge that socioeconomic processes and their governing principles (e.g., policies, cultural preferences) can change drastically over space and time, existing spatiotemporal models of socioeconomic processes have conventionally focused on short-term futures and/or small geographic areas, and have shown strong limitations for long-term, large-scale investigations (e.g., those discussed above regarding spatial population projections). New developments are needed in this realm, and the purpose of this kind of modeling is not to predict exactly what will happen in the far future, but rather to understand potential long-term effects of alternative socioeconomic choices. The forward thinking tools developed for this purpose can also help address other deep-rooted societal issues (e.g., environmental justice).

Finally, urbanization is a natural outcome of social and economic developments. Instead of seeing it as an unwelcome trend, it may be more productive to view it as an ongoing process, and as such, it is within our power to decide how urban patterns should and will be arranged. Climate change is a long-term, large-scale challenge, but if through addressing it, we learn to incorporate long-term planning in

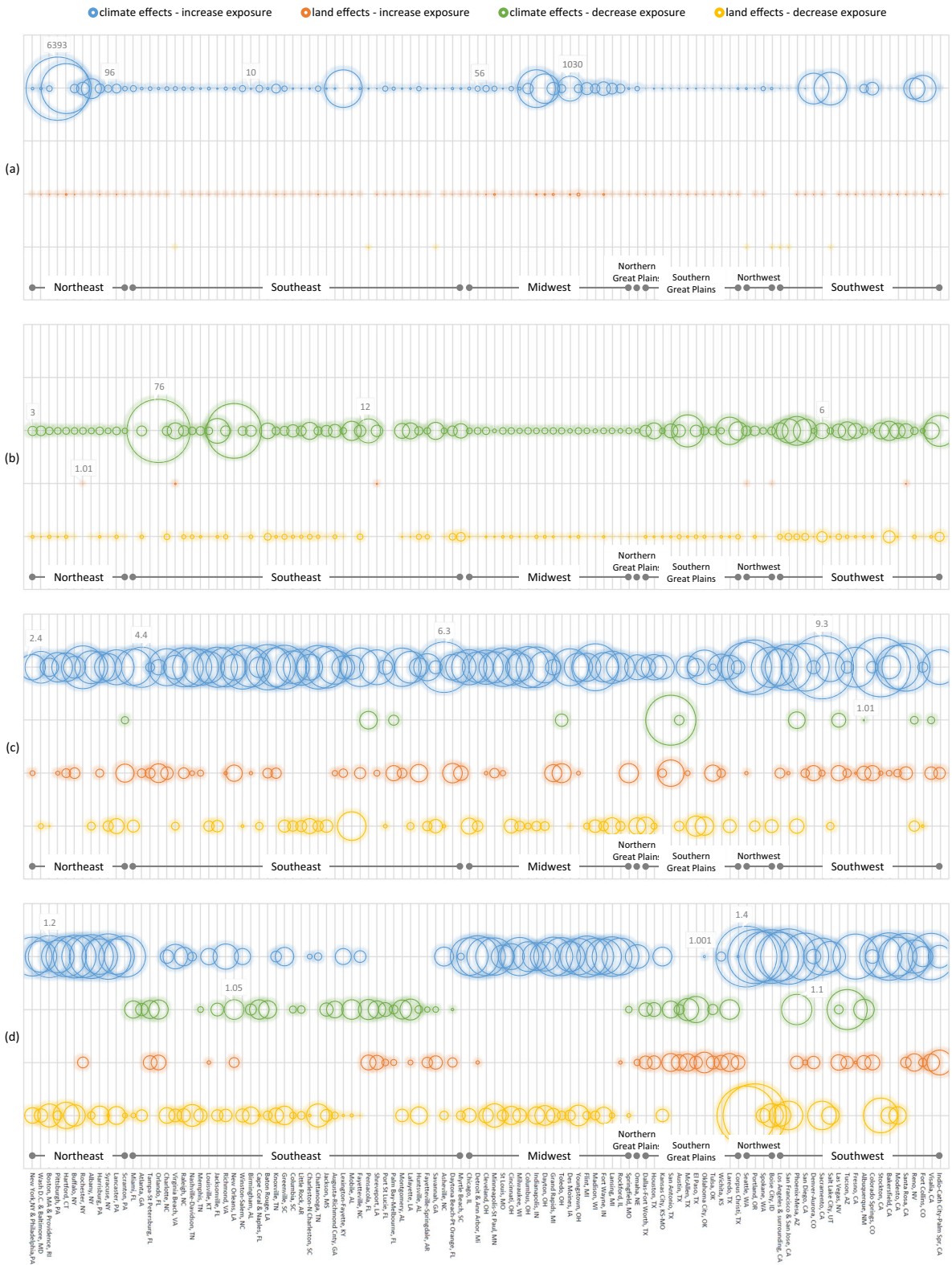

**Fig. 4 | Effects of climate change and urban land expansion from the beginning (BOC) to the end (EOC) of the 21st century on population exposures in 109 sizeable urban centers to four climate extremes. a** hot days, **b** cold days, **c** heavy rainfalls, and **d** severe thunderstorm environments. The urban centers are grouped by climate regions, and within each region are ordered by descending population size from left to right. Effects are measured as ratios: When an effect increases exposure, it is reported as $\left(\frac{exposure_{EOC}}{exposure_{BOC}}\right)$; when an effect decreases exposure, it is reported as $\left(\frac{exposure_{BOC}}{exposure_{EOC}}\right)$. Super large ratios are usually results of very small denominators rather than extraordinarily large numerators. Areas of the circles are scaled in proportion to sizes of effects within each panel, and benchmark circle sizes are labeled for each panel. That is, sizes of circles can be compared within each panel, but not across panels.

our policy- and decision-making, and better collaborate at large scales, the challenge can turn into an opportunity for societal growth.

## Methods

For climate projections, we use a set of published scenarios from our recent work[8,29], which were produced using the Weather Research and Forecasting (WRF) model at a 25-km resolution for the continental U.S. The beginning of the century (BOC) conditions were reported as 1980–2005 summaries, and the end of the century (EOC) conditions 2075–2100 summaries. Two EOC climate scenarios were considered: (1) reflecting only climate change under the Representative Concentration Pathway[30] (RCP) 8.5, and (2) reflecting both RCP 8.5 climate change and the Shared Socioeconomic Pathway[31] (SSP) 5 land-use change impacts on climate conditions. Both urban and agricultural land changes (including crop and pasture lands) were incorporated[25,32]. However, since SSP 5 is high in urban land change and minimal in agricultural land change for the U.S., we refer to the scenario as a high urban land expansion scenario.

The spatial urban land projections (SI Fig. 1) were produced by a new data-science-based urban land modeling framework[4,25,26] (namely, CLUBS-SELECT) which draws on 15 best available datasets on contemporary urbanization and its driving forces, and outperforms previous urban land models by reflecting (i) changes in urbanization styles over long time horizons, (ii) local-scale spatial variations in the urbanization process while maintaining global coverage (by modeling the world as 375 sub-national regions, each with one unique spatial model), and (iii) the emergence of new urban centers (in contrast to previous urban land models that generally expand existing urban centers for new development).

Using these climate projections, four weather extremes were defined for each 25-km grid (SI Fig. 2), representing a variety of potential risks to human wellbeing and infrastructure: A hot day is when the daily maximum temperature is greater than or equal to 35 °C, a cold day is when the daily average temperature is less than or equal to 0 °C, a heavy rainfall is a precipitation event whose hourly precipitation amount is greater than or equal to the hourly precipitation amount of a 5-year return period storm at BOC calculated for every grid in the U.S. respectively, and a severe thunderstorm environment (STE) day is when the daily product of the Convective Available Potential Energy (CAPE) and the 0–6 km wind shear is greater than 20,000 $m^3s^{-3}$. Our hot extreme definition is commonly used in population exposure assessment for the U.S.[5] For cold extreme definitions, no widely-accepted metric exists in climate or related mortality research[33], and we choose a severe and relatable threshold for infrastructural impacts (e.g., freezing pipes and roads). Our precipitation extreme definition is based on literature pertinent for urban flooding and stormwater management[34–36], and five-year return periods can be reliably calculated from our climate projections. The STE definition[37] marks days with large-scale atmospheric environments that may form hazardous convective weather (i.e., thunderstorms, tornadoes, large hails, damaging wind gusts) potentially leading to severe human and infrastructural loss[38].

Exposures were calculated at the 25-km resolution, for BOC and EOC respectively, by taking the product of the BOC or EOC average frequency map of each weather extreme with the 2005 or 2090 spatial population map (SI Fig. 1), which are aggregated from 1-km spatial population data[39,40] to show the total population count within each 25-km grid.

Different aggregation strata were applied to the exposure maps to help understand the spatial patterns of exposures. For urban development density categories, the "not developed" category consists of grids with less than 20% urban land, the "low-density development" category 20–50%, and "mid-to-high-density development" greater than 50%. For climate regions, we use the U.S. National Climate Assessment (NCA) definitions[41]. For sizeable urban centers, we use

latitude and longitude coordinates from the United Nations' World Urbanization Prospects[24] to locate the 143 sizeable cities in the continental U.S. that have a population size greater than 300,000. Then, we create 20-km buffers for cities whose population sizes are between 300 and 500 thousand, 25-km buffers for cities with population sizes between 500 thousand and 1 million, 50-km buffers for 1–5 million, 75-km buffers for 5–10 million, and 100-km buffers for cities whose population sizes are greater than 10 million. Resolving merging boundaries of the buffers, we have 109 urban centers across the continental U.S. (SI Fig. 3). The buffers centered around the latitude and longitude coordinates of cities take circular shapes. While no city is exactly a circle, this analytical choice is deliberate: Because the purpose of this analysis is to examine, at the regional scale, how climate conditions interact with urban land systems in their regional contexts, we need a boundary definition for land systems that centers around urban areas while also includes relevant regional surroundings, rather than a definition of urban extent only. Furthermore, while urban extents change over time, for the purpose of this analysis, the boundary of a land system should stay the same for BOC and EOC, so that all effects of urban land expansion can be captured in the results. If a spatial definition of changing urban extent had been applied, the changing extent would have accounted for some urban land expansion effects (e.g., those from increasing the urban fraction of a land system) and therefore exclude such effects from showing in results. For example, imagine a land system that consists of 5 urban grids and 5 agricultural grids at BOC, and by EOC all 10 grids are urban. When comparing temperatures averaged over the 10 grids for BOC and EOC (i.e., using a fixed boundary for the 10-grid land system), effects of converting agricultural lands for urban uses are captured through the 5 grids that are agricultural at BOC, and effects of urban expansion on existing urban lands are captured through the 5 grids that are already urban at BOC. But, if average temperature is calculated using 5 urban grids for BOC and 10 urban grids for EOC (i.e., using changing boundaries of urban extent), effects of converting agricultural lands for urban uses (an important effect of urban land expantion) will be excluded from the results. Hence, though circular buffers might seem an unusual choice for defining urban areas, it suits the need of this research to define regional land systems encompassing urban areas. Different buffer radiuses are applied to different city population size categories, as cities with larger population sizes occupy more land areas and interact with larger regions. Using a uniform radius would either cut off parts of large cities, or dilute effects of small cities with too much regional contexts of little land change. The specific radius for each city population size category was chosen so that urban centers in the category will be completely encompassed by their buffers at EOC, i.e., after the maximum urban land expansion has taken place.

Four key scenarios, combining different climate, land use, and population conditions, exist in our dataset (SI Table 1). The "historical" scenario shows exposures under the BOC historical conditions of all three variables. The "climate" scenario shows EOC exposures considering RCP 8.5 climate change over the 21st century, with no land use nor population change. The "climate+pop" scenario is the current common practice, overlaying projected future spatial population and climate conditions (which reflect only climate change effects and assume current land-use patterns hold constant throughout the 21st century). The "climate+pop+landUse" scenario reflects effects of projected changes over the 21st century in all three variables (climate, land use, population). RCP 8.5 and SSP 5 are both high change scenarios. We recognize that high scenarios may not be the most likely scenarios, while their high signal-to-noise ratios and ability to offer more data points on climate extremes can help the comparison of relative importance of effects from different climatic and social changes.

To isolate effects on population exposure to climate extremes from the three variables respectively (climate, land use, population), we compare the stratified aggregate exposure counts (ECs) from the

four above mentioned scenarios with each other (SI Fig. 4). In theory, because exposures are defined as the product of population size and average frequency of each climate extreme, ratio rather than subtraction based metrics would more naturally separate effects from climate, land use, and population changes, while in practice, subtraction based metrics are broadly used. Here we test both kinds of effect metrics. In this research, multiple different pairs of scenarios may be used to measure effects of changes in the same variable. For example, for measuring land-use change effects, the scenario pairs (climate+pop+landUse, climate+pop) and (climate+landUse, climate) are both conceptually appropriate. However, when using subtraction based metrics, $(EC_{\text{climate+pop+landUse}} - EC_{\text{climate+pop}})$ and $(EC_{\text{climate+landUse}} - EC_{\text{climate}})$ result in largely different values, e.g., for the national total exposure to hot days, $(EC_{\text{climate+pop+landUse}} - EC_{\text{climate+pop}})$ is 5.9 billion person days, and $(EC_{\text{climate+landUse}} - EC_{\text{climate}})$ is 2.4 billion person days; when using ratio based metrics, $(EC_{\text{climate+pop+landUse}}/EC_{\text{climate+pop}})$ and $(EC_{\text{climate+landUse}}/EC_{\text{climate}})$ result in nearly identical values, e.g., also for the national total exposure to hot days, $(EC_{\text{climate+pop+landUse}}/EC_{\text{climate+pop}})$ is 1.28, and $(EC_{\text{climate+landUse}}/EC_{\text{climate}})$ is 1.27. This finding is consistent with the theoretical reasoning that for multiplication-based metrics (e.g., exposure counts), ratio based metrics can clearly separate effects from different amplifying factors. Essentially, the fact that $(EC_{\text{climate+pop+landUse}} - EC_{\text{climate+pop}})$ is much larger than $(EC_{\text{climate+landUse}} - EC_{\text{climate}})$ is because the former captured some population effects while aiming to capture only urban land effects. To confirm the robustness of ratio based effect measures broadly holds, we generate two helper scenarios: "climate+landUse" (capturing effects of changes in both climate and land use, but assuming no population change), and "pop" (combining historical climate and land use with future population conditions), and used all possible scenario pairs for calculating climate, land use, and population effects across all national totals, urban development densities, climate regions, and sizeable urban centers. The results show, empirically, the robustness of the effect ratios always holds. Therefore, we finalize our definitions of effects using only the four key scenarios, and define climate effects as $(EC_{\text{climate}}/EC_{\text{historical}})$, land-use effects as $(EC_{\text{climate+pop+landUse}}/EC_{\text{climate+pop}})$, and population effects as $(EC_{\text{climate+pop}}/EC_{\text{climate}})$.

The effect ratio approach also works well for separating climatic and land-use change effects on climate conditions, i.e., the endpoint variable does not have to be ECs. For example, across sizeable U.S. urban centers, climate effects on annual total precipitation and summer total precipitation (June, July, and August) are both around a factor of 1.1, while climate effects on heavy rainfalls are more than 3 (SI Fig. 5). That is, although the total amount of precipitation in U.S. urban centers does not change much over the 21st century, a substantially higher percentage of precipitation will occur as heavy rainfalls.

While using state-of-the-art methods, this work is still affected by some methodological limitations, primarily: First, the climate simulations used only one model, configuration, and forcing combination. Though a well-documented, well-regarded, and widely-used choice, further research examining the effects of model structural uncertainties on studying society-climate interactions is important. A detailed discussion about potential future research in this direction can be found in our recent paper[8]. Second, due to the insufficiencies of current long-term spatial population models and projections discussed earlier, new model development is needed, before in-depth examinations of how spatial population patterns interact with urban land and climate can be possible. We are currently working on these research directions.

## Data availability
Source Data underlying all figures are provided as a Supplementary Dataset, with one spreadsheet per figure. The spatial projections of urban land[25,32], population[39,40], and climate conditions[8,29] used in this research are publicly available as referenced. Intermediate materials are available upon request. Source data are provided with this paper.

## Code availability
Codes used to produce this work are available upon request.

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

## Acknowledgements

This research has been partly supported by the U.S. National Science Foundation (NSF) [grants 1243095 & 2239859 for JG; major facility cooperative agreement 1852911 (the U.S. National Center for Atmospheric Research, NCAR) for MSB] and the U.S. Department of Energy (DOE) [grants DE-SC0016438 & DE-SC0016605 for MSB]. MSB completed this research while employed at NCAR in the Research Applications Laboratory (RAL). She would like to acknowledge NCAR/RAL for their support, as well as NCAR's Computational and Information Systems Laboratory (CISL) for providing computing support and resources (the Casper data analysis and visualization cluster, doi:10.5065/D6RX99HX), sponsored by NSF. The authors would like to thank Lee Kessenich at NCAR for their assistance with some early analyses.

## Author contributions

J.G. and M.S.B. collaborated equally to conceive, design, and implement this research.

## Competing interests

The authors declare no competing interests.
