## [Peer Review File · Nature Communications]

Urban Land Patterns Can Moderate Population Exposures to Climate Extremes over the 21st CenturyREVIEWER COMMENTS

Reviewer #1 (Remarks to the Author):

The manuscript analyzes the impacts of climate change and urbanization on the exposure of the US population to extreme weather events (particularly heat, cold temperatures, precipitation, and storms) by the end of the century. The focus is on the impact of land cover change and how excluding this variable in climate models (as often the case) can lead to underestimating human exposure to extreme weather events. The methodology draws on a number of existing models developed by the authors and their peers, but the paper has a distinct enough research question compared to these previous works.

However, while this topic is of interest to a large audience, I have some key concerns with the methodologies and discussion thereof as well as several minor comments.

Key concerns:

1) The manuscript spends most of its time describing symptoms and gives no information on the sources of these symptoms. I.e., the manuscript lacks any discussion on the physical mechanisms that make the impact of land cover on weather events so important. While I understand that the physical nature of climate is not the topic and there is a limit in the number of words, an interdisciplinary audience might not be familiar with the links between land cover changes and climate.

An example of this would be in line 101 where they note that the high urbanization scenario results in less heavy rainfall events in densely populated regions (whereas it has no effect on unpopulated lands and increases rainfall extremes in low density development – this is only shown in the supplement). This kind of leaves me wondering why heavy rainfall decreases in areas of high urbanization but increases in areas with low urbanization.

2) The justification of the authors for using ratios as the metric for comparison is very weak. There is again no discussion of the physical mechanics of urbanization on extreme weather events and if we would expect those to add to the effects of climate change (i.e., subtraction for comparison) or amplifying them (i.e., ratio for comparison). The way it is currently written makes it seem the authors tried a bunch of stuff and this one gave the best results. However, we do not get to see what they have tried, and we get no explanation why they think it works, leaving me wondering if this is a case of 'correlation does not equal causation'

3) I am very dubious about the authors analysis of exposure in urban centers.

i) the assumptions that all cities are a circle is very much wrong and there are databases out there that map city clusters. I would very much recommend using e.g., the Global Human Settlement Layer (<https://ghsl.jrc.ec.europa.eu/ucdb2018Overview.php>)

ii) With increasing population the area of these cities will likely change, are your projects for land use and your city boundaries compatible?

iii) I do not agree that identifying three cities where urbanization will slow down exposure to weather extremes is a novel or even all that exciting find. Urban cooling particularly in arid regions (where these three cities are located if I read the figure correctly – but it's pretty hard to read...) is not a new phenomenon and (in part) due to higher evapotranspiration rates in these cities compared to the arid regions outside. I hence do not feel this potentially not even significant changes that you modeled for three cities warrants the attention you give them in discussion and abstract.

4) It is my understanding that CMIP6 in a way combines the SSPs and RCP scenarios – If this is the case (and it might not be, this isn't my area of expertise) I would be very curious if the authors feel these new standard climate scenarios do a much better job of getting the everyday user to include land cover change in their analysis?

Detailed comments:

1. Title and throughout the manuscript: The term 'population experience' sounds strange to me and I couldn't find evidence of this term being used regularly in this field. I'm not a native speaker but I'm

wondering if 'exposure' would be more appropriate.

2. Line 52 and following: I'd suggest to not mention the denotations (Climate_landUse_popular conditions) in the introduction or even anywhere in the manuscript. They are barely used anyways, and it might be easier for the reader to just name the different scenarios as: Climate, Climate+population, or climate+landUse etc. As you are only using RCP 8.5 and SSP5 we do not really highlight that information.
3. Table S1 - I'm a bit confused about the unit for the per capita exposure. 'Days per person per year' - why per person? This would mean that in an area of 1000 people and a per capita exposure of 30 the year has 30 days person-1 years-1*1000 persons = 30000 days per year ...
4. Line 71 to 74 - how did you come to this conclusion?
5. Line 91 - is this not primarily an effect of ongoing urbanization, i.e, a higher percentage of people living in cities?
6. Line 94 - mention that this is the case for all land uses/degrees of urbanization
7. Line 96 and following - if you do introduce denotations for your models now would be a good time to use them.
8. Fig 1 - I would suggest showing percentages instead of what you are doing now. (i.e., what is now 1 or -1 would become 100%). Might make things easier to read.
9. Line 134 and following- how do these findings compare to other studies modelling exposure to extreme weather events?
10. Fig 2 Like with Fig 1 I suggest you use percentages.
11. Fig 2 what models are you using to model climate effects and urban land use effects
12. Line 181 please rephrase. 'First, climate.' Is not a sentence.
13. Fig 3 - units missing and y-axis descriptors missing in figure
14. Fig 4 - text too small, unreadable.
15. Line 290 missing 'as' before 2075
16. Fig S1 units missing (population per km²?)
17. Line 297 following. What's the goodness of their models. While I understand that these models have been validated in previous studies, I feel some sort of discussion warranted. Also, what input parameters specifically are used from the SSP and RCP scenarios?
18. Fig S2 - hard to compare as they are so similar. (Outside of the urban heat islands in the first row)
19. Line 310 where did you get data on a 5-year return period storm. Do BOC and EOC have different thresholds as the 5-year return period storm changes?
20. Fig S4 - legend (circle size) missing
21. Around Line 352 and following - the figure S4 only shows 3 scenarios, not the five you discuss here

Reviewer #2 (Remarks to the Author):

Dear Drs. Gao and Bukovsky,

The paper entitled, "Urban land patterns can moderate population experiences of 21st-century climate change," explores the effect of urbanization and climate change on population exposure to hot days, cold days, rainfall, and severe storms in urbanized areas of the United States. Overall, I find this work to be important, timely, and relevant to the journal. I thank the authors for a well structured manuscript with methods that are easy to follow.

I have no major comments at this time. The methods are sound and the results are appropriately interpreted. There is some clarification needed, as explained below, but this is very minor.

The Introduction, specifically focusing on literature review, is very short and seems to be missing

information.

For example, on lines 26-27, you mention that urbanization increases population exposure. It would be extremely helpful to discuss why this is the case as you then discuss modifying land use in further parts of the introduction. What about urban land use specifically causes these patterns of exposure?

Lines 141-142: While I agree that some areas will increase in vulnerability because of lack of acclimatization, past experiences, and protective measures, do you consider the southeast in this position? Much of the SE US has dealt with heat stress, especially when including humidity, every summer.

Line 161: What societal factors would specifically cause less exposure?

Lines 175-177: You mention later in the manuscript that this is an area for future research, but is there anything you can generally suggest for how urban land can be used to moderate climate extremes?

Reviewer #3 (Remarks to the Author):

Review: Urban Land Patterns Can Moderate Population Experiences of 21st-1 Century Climate Extremes

General comments: This is a paper that will likely be of great interest to readers. The objective is to examine how changes in urban land, population, and climate will influence spatial patterns of population exposure to climate stressors (cold and hot temp extremes, heavy precip, and severe thunderstorm events) in the future. Results diverge from what's commonly perceived in the field in that urban land can in some cases reduce population exposure to climate extremes, including hot temp extremes. Further, findings indicate that spatial patterns rather than total quantity of urban land played a larger role. One potential limitation is the validity of the effect ratio approach. While I'm struck by the simple elegance of this measure, the authors don't currently reference others in the field who have used this approach to tease out land use from climatic changes. However, it appears that the effect ratio is only measuring the relative contribution of climate change or urban land patterns on population exposure. Why not take the study a step further and incorporate an absolute measure that shows the risk difference (relative and absolute measure) that shows the actual number for population exposed to quantify the excess contribution of climate change or urban land on population exposure to climate extremes. The methods section could benefit from further sharpening. There is a great deal of study required for the paper involving keeping track of the various scenarios, spatial aggregation scales, and climate models. More fine-tuning to enhance readability in the naming conventions of the various scenarios would be helpful.

Recommendation: Major revisions

Introduction

It's not clear that there has been a strong climate signal implicated in severe thunderstorms and wondered why the authors decided to include this in the line-up of extremes rather than say drought events as a compliment to heavy precip? Is it more of a conceptual control that removes climate changes and just looks at the influence of population and land use changes (only)?

A text box or table mapping the different scenarios used might provide a helpful quick reference for the reader

Results

Should there be a table or figure referenced in the first couple of paragraphs of the Results section?

The authors state: "We capture effects of changes in climate, urban land, and population over the 21st century as ratios between exposure counts (ECs) from different scenarios, because such ratios can robustly separate effects of climate, urban land, and population (more explanations in Methods). More rational as to how and why these ratio measures adequately measure the effects of climate change from BOC to EOC is needed

BOC to EOC on population exposure to a climate 78 extreme as (*ECRCP8.5_hist_hist/ EChist_hist_hist*), effects of urban land expansion as 79 (*ECRCP8.5_SSP5_SSP5/ ECRCP8.5_hist_SSP5*), and effects of population change as (*ECRCP8.5_hist_SSP5/80 ECRCP8.5_hist_hist*).

SI Table 1. Can the authors better orient the reader for the inclusion of Total Exposure and Per Capita Exposure and how they arrive at billion person days vs days per person per year? Is the table meant to reflect total change in population exposure between the two exposure scenarios and across climate extremes??? A range would be more helpful in reporting (e.g., hot days is not developed areas by end of century 26% (18, 32)

Re: stationarity in population change estimates lending to insufficiencies in analyzing potential changes in spatial patterns - pg 6 line 116?

Fig 2 -can the authors comment on the large ratios for the climate effects and hot days 27.56, 21.21

Fig 3 conveys a large amount of information, but takes a lot of studying for the reader to be able to derive meaning. Wonder if there might be a more effective way to visualize these results.

Especially, the 214 urban land projections we used in this research are produced by a novel data-driven model^{4,24,25} 215 (more information in Methods). [incomplete?]

Fig 4 - nice and effective visual

Discussion

Lacks a strengths/limitations section

Methods

Is there an inherent benefit/advantage of using the Weather Research and Forecasting (WRF) model?

There are a number of absolute and percentile-based thresholds for defining extreme temperature events (e.g., above the 95th or below the 5th percentile based on the climatology of a given geographic area). The justification for the cold and hot temp days is not well substantiated by the literature and why not include a couple of additional metrics in a sensitivity analysis?

Re: different aggregation strata for exposure maps. How did the authors arrive at the 20-KM - 100-km buffers for cities with a population larger than 300k? There still seems to be spatial overlap among large cities in the Northeast and out west in California.

Re: scenarios - for the different scenarios - can the authors come up with a better naming convention to aid interpretability for the reader, there are a lot of acronyms to keep track with that are not intuitive (RCP8.5_SSP5_SSP5, RCP8.5_hist_SSP5, RCP8.5_SSP5_hist, RCP8.5_hist_hist) and then spell out these scenarios in figures/tables + captions?

Re: effect ratios - it's not clear if this is a standard approach already used in the field to compare exposure count climate scenarios.

Re: To confirm the robustness of measuring effects as ratios, we added a dummy scenario 362

hist_hist_SSP5 to our scenario pool, and used all possible scenario pairs for calculating climate, 363 land use, and population effects as ratios across all national totals, urban development densities, 364 climate regions, and sizeable urban centers. The robustness of the effect ratios always holds. Can the authors point the reader to these results in supplemental materials? There are no citations providing justification of this approach.

Dear reviewers,

Thank you for taking the time to help us improve this manuscript! Below we address your comments one by one. Hope you find the changes satisfactory.

Best regards,

Jing Gao & Melissa Bukovsky

Reviewer #1:

The manuscript analyzes the impacts of climate change and urbanization on the exposure of the US population to extreme weather events (particularly heat, cold temperatures, precipitation, and storms) by the end of the century. The focus is on the impact of land cover change and how excluding this variable in climate models (as often the case) can lead to underestimating human exposure to extreme weather events. The methodology draws on a number of existing models developed by the authors and their peers, but the paper has a distinct enough research question compared to these previous works.

However, while this topic is of interest to a large audience, I have some key concerns with the methodologies and discussion thereof as well as several minor comments.

Thank you for helping us improve this work.

Key concerns:

1) The manuscript spends most of its time describing symptoms and gives no information on the sources of these symptoms. I.e., the manuscript lacks any discussion on the physical mechanisms that make the impact of land cover on weather events so important. While I understand that the physical nature of climate is not the topic and there is a limit in the number of words, an interdisciplinary audience might not be familiar with the links between land cover changes and climate.

An example of this would be in line 101 where they note that the high urbanization scenario results in less heavy rainfall events in densely populated regions (whereas it has no effect on unpopulated lands and increases rainfall extremes in low density development – this is only

shown in the supplement). This kind of leaves me wondering why heavy rainfall decreases in areas of high urbanization but increases in areas with low urbanization.

Edits: Added new text about relevant physical mechanisms in the Introduction section, at lines 50-57: “Existing literature on future population’s vulnerability to climate extremes has focused on temperature^{7,15,16}. However, patterns of extreme precipitation and convective environments will also evolve, as both climate change and urbanization modify future land-atmosphere dynamics¹⁷⁻²². Effects of urban heat island (UHI) not only strengthen sensible heat fluxes to increase temperature, but also enhance surface roughness and atmospheric circulations to influence the frequency and the intensity of severe convective storms²³, including extreme precipitation and thunderstorms. Here we examine hot days, cold days, heavy rainfalls, and STEs.”

Added explanations of the example at line 101 (now lines 110-122): “Surprisingly, urban land expansion can also decrease average exposures to climate extremes in some cases. For example (SI Table 2), urban land expansion in mid-to-high-density developments decreases the EOC per capita exposure to heavy rainfalls. This change in exposure might be caused by physical land-atmosphere mechanisms (e.g., UHI effects can strengthen convective inhibition over densely developed urban areas⁸, which can decrease the frequency of extreme precipitation events to counteract the broadly-occurring enhancing effect of climate change), and/or spatial population-urban-land dynamics (e.g., many densely developed urban areas are not residential and hence show low population exposures). Identifying what patterns and mechanisms can reduce population exposures to climate extremes, and using that knowledge to inform urban designs, are our planned next research steps. The significance of this work is that the results show carefully designed urban land patterns are able to practically moderate instead of intensify the nation’s future experiences of climate extremes (more evidence in later sections).”

Added discussion about future research plans, at lines 288-292: “It is therefore worthwhile and beneficial to identify urban land patterns and related physical and socioeconomic mechanisms that can reduce population exposures to climate extremes over the long term, and use such understandings to guide urban designs for the future.”

2) The justification of the authors for using ratios as the metric for comparison is very weak. There is again no discussion of the physical mechanics of urbanization on extreme weather events and if we would expect those to add to the effects of climate change (i.e., subtraction for comparison) or amplifying them (i.e., ratio for comparison). The way it is currently written makes it seem the authors tried a bunch of stuff and this one gave the best results. However, we do not get to see what they have tried, and we get no explanation why they think it works, leaving me wondering if this is a case of ‘correlation does not equal causation’

Edits: Added explanations explicitly reasoning the choice of using ratios rather than subtractions for separating climate, land use, and population effects, and added one numerical example, at lines 417-451: “To isolate effects on population exposure to climate extremes from the three variables respectively (climate, land use, population), we compare the stratified aggregate exposure counts (ECs) from the four above mentioned scenarios with each other (SI Fig 4). In theory, because exposures are defined as the product of population size and average frequency of each climate extreme, ratio rather than subtraction based metrics would more naturally separate effects from climate, land use, and population changes, while in practice, subtraction based metrics are broadly used. Here we tested both kinds of effect metrics. In this research, multiple different pairs of scenarios may be used to measure effects of changes in the same variable. For example, for measuring land use change effects, the scenario pairs (climate+pop+landUse, climate+pop) and (climate+landUse, climate) are both conceptually appropriate. However, when using subtraction based metrics, $(EC_{\text{climate+pop+landUse}} - EC_{\text{climate+pop}})$ and $(EC_{\text{climate+landUse}} - EC_{\text{climate}})$ result in largely different values, e.g., for the national total exposure to hot days, $(EC_{\text{climate+pop+landUse}} - EC_{\text{climate+pop}})$ is 5.9 billion person days, and $(EC_{\text{climate+landUse}} - EC_{\text{climate}})$ is 2.4 billion person days; when using ratio based metrics, $(EC_{\text{climate+pop+landUse}} / EC_{\text{climate+pop}})$ and $(EC_{\text{climate+landUse}} / EC_{\text{climate}})$ result in nearly identical values, e.g., also for the national total exposure to hot days, $(EC_{\text{climate+pop+landUse}} / EC_{\text{climate+pop}})$ is 1.28, and $(EC_{\text{climate+landUse}} / EC_{\text{climate}})$ is 1.27. This finding is consistent with the theoretical reasoning that for multiplication-based metrics (e.g., exposure counts), ratio based metrics can clearly separate effects from different amplifying factors. Essentially, the fact that $(EC_{\text{climate+pop+landUse}} -$

$EC_{\text{climate+pop}}$) is much larger than $(EC_{\text{climate+landUse}} - EC_{\text{climate}})$ is because the former captured some population effects while aiming to capture only urban land effects. To confirm the robustness of ratio based effect measures broadly holds, we generated two helper scenarios: “climate+landUse” (capturing effects of changes in both climate and land use, but assuming no population change), and “pop” (combining historical climate and land use with future population conditions), and used all possible scenario pairs for calculating climate, land use, and population effects across all national totals, urban development densities, climate regions, and sizeable urban centers. The results show, empirically, the robustness of the effect ratios always holds. Therefore, we finalize our definitions of effects using only the four key scenarios, and define climate effects as $(EC_{\text{climate}} / EC_{\text{historical}})$, land use effects as $(EC_{\text{climate+pop+landUse}} / EC_{\text{climate+pop}})$, and population effects as $(EC_{\text{climate+pop}} / EC_{\text{climate}})$.”

3) I am very dubious about the authors analysis of exposure in urban centers.

i) the assumptions that all cities are a circle is very much wrong and there are databases out there that map city clusters. I would very much recommend using e.g., the Global Human Settlement Layer (<https://ghsl.jrc.ec.europa.eu/ucdb2018Overview.php>)

ii) With increasing population the area of these cities will likely change, are your projects for land use and your city boundaries compatible?

iii) I do not agree that identifying three cities where urbanization will slow down exposure to weather extremes is a novel or even all that exciting find. Urban cooling particularly in arid regions (where these three cities are located if I read the figure correctly – but it’s pretty hard to read...) is not a new phenomenon and (in part) due to higher evapotranspiration rates in these cities compared to the arid regions outside. I hence to not feel this potentially not even significant changes that you modeled for three cities warrants the attention you give them in discussion and abstract.

Edits: For points (i) and (ii), added clarifications in the Methods section, at lines 367-401: “For sizeable urban centers, we used latitude and longitude coordinates from the United Nations’ World Urbanization Prospects²⁴ to locate the 143 sizeable cities in the continental U.S. that have a population size greater than 300,000. Then, we created 20-km buffers for

cities whose population sizes are between 300 and 500 thousand, 25-km buffers for cities with population sizes between 500 thousand and 1 million, 50-km buffers for 1-5 million, 75-km buffers for 5-10 million, and 100-km buffers for cities whose population sizes are greater than 10 million. Resolving merging boundaries of the buffers, we have 109 urban centers across the continental U.S. (SI Fig 3). The buffers centered around the latitude and longitude coordinates of cities take circular shapes. While no city is exactly a circle, this analytical choice is deliberate: Because the purpose of this analysis is to examine, at the regional scale, how climate conditions interact with urban land systems in their regional contexts, we need a boundary definition for land systems that centers around urban areas while also includes relevant regional surroundings, rather than a definition of urban extent only. Furthermore, while urban extents change over time, for the purpose of this analysis, the boundary of a land system should stay the same for BOC and EOC, so that all effects of urban land expansion can be captured in the results. If a spatial definition of changing urban extent had been applied, the changing extent would have accounted for some urban land expansion effects (e.g., those from increasing the urban fraction of a land system) and therefore exclude such effects from showing in results. For example, imagine a land system that consists of 5 urban grids and 5 agricultural grids at BOC, and by EOC all 10 grids are urban. When comparing temperatures averaged over the 10 grids for BOC and EOC (i.e., using a fixed boundary for the 10-grid land system), effects of converting agricultural lands for urban uses are captured through the 5 grids that are agricultural at BOC, and effects of urban expansion on existing urban lands are captured through the 5 grids that are already urban at BOC. But, if average temperature is calculated using 5 urban grids for BOC and 10 urban grids for EOC (i.e., using changing boundaries of urban extent), effects of converting agricultural lands for urban uses (an important effect of urban land expansion) will be excluded from the results. Hence, though circular buffers might seem an unusual choice for defining urban areas, it suits the need of this research to define regional land systems encompassing urban areas. Different buffer radiuses are applied to different city population size categories, as cities with larger population sizes occupy more land areas and interact with larger regions. Using a uniform radius would either cut off parts of large cities, or dilute effects of small cities with too much regional contexts of little land change. The specific radius for each city population size category was chosen so that urban centers

in the category will be completely encompassed by their buffers at EOC, i.e., after the maximum urban land expansion has taken place.”

For point (iii), added clarifications about the finding in the Results section, at lines 230-242: “For temperature extremes, urban land expansion generally leads to warming, due to UHI effects, but exceptions exist for a few urban centers, to reduce exposure to hot days and increase exposure to cold days. Though these exceptions are of small magnitudes, their existence showcases the possibility of using urban land effects to moderate, or at least not horribly amplify, climate effects. Especially for hot days, a total of seven urban centers’ land changes reduce exposures, i.e., Savannah, GA, Los Angeles, CA, Boise, ID, Seattle, WA, San Francisco, CA, Salt Lake City, UT, and Virginia Beach, VA, with their urban land effects on population exposures to hot days ranging from 0.92 to 0.48. This is NOT to say that the current urban land patterns of these cities already moderate population exposures, but rather their urban land changes simulated in our hypothesized high urbanization scenario show that effect. Investigations of what mechanisms drive the exposure reductions are planned for our next research. We will consider both physical factors (e.g., changes in evapotranspiration, albedo, sea breeze) and population-land dynamics (e.g., infill development, migration).”

Also for point (iii), this is a bit too technical to include in the main text of the paper: It is true that higher evapotranspiration due to irrigated landscaping can lead to cooling for urban areas in arid regions. However, for Salt Lake City (the only arid city out of the seven), in our simulations, that mechanism isn’t the cause of exposure reduction, because in the WRF configuration we used, higher evapotranspiration with urbanization would only occur if urban land replaced barren land, which is not the case here. In our next research, we will look into what physical factors (including evapotranspiration) led to the exposure reduction. Hope you will stay tuned to our future findings.

4) It is my understanding that CMIP6 in a way combines the SSPs and RCP scenarios – If this is the case (and it might not be, this isn’t my area of expertise) I would be very curious if the authors feel these new standard climate scenarios do a much better job of getting the everyday user to include land cover change in their analysis?

Edits: You are right that CMIP6 made a greater effort incorporating SSP information than previous versions, but important limitations remain (e.g., the land cover data used by CMIP6 do not reflect the current state of the art for land cover change modeling, especially for urban land for long term). For “everyday users”, we still feel more demonstration and broadcasting are needed to help people see geophysical ramifications of human choices about land use/cover, to inspire more commitments to actions. We’d like to think our works, along with efforts from many others, are helping make the case for better considering land cover change in regional and global climate modeling and planning.

Detailed comments:

1. Title and throughout the manuscript: The term ‘population experience’ sounds strange to me and I couldn’t find evidence of this term being used regularly in this field. I’m not a native speaker but I’m wondering if ‘exposure’ would be more appropriate.

Edits: Changed to “exposure” throughout the manuscript including the title, at lines 1, 11, 21, 28, 34, 38, 49, 174, 227, 247, 256, 270, 275.

2. Line 52 and following: I’d suggest to not mention the denotations (Climate_landUse_popular conditions) in the introduction or even anywhere in the manuscript. They are barely used anyways, and it might be easier for the reader to just name the different scenarios as: Climate, Climate+population, or climate+landUse etc. As you are only using RCP 8.5 and SSP5 we do not really highlight that information.

Edits: Changed the notations of the scenarios throughout the manuscript as recommended, including texts, figures, tables, SI, etc. For example, at lines 58-72: “We analyzed three future scenarios combining different climate, land use, and population conditions at the end of the 21st century (EOC; 2075-2100 summaries) for spatial population exposures to the four climate extremes at a 25-km resolution (more information in Methods and SI Table 1): (i) high greenhouse gas warming with no population nor urban land change (denoted as “climate”, showing effects from climate change only), (ii) high greenhouse gas warming and high population growth with no urban land change (“climate+pop”, the current common practice), and (iii) high greenhouse gas warming, high population growth, and high urban land expansion (“climate+pop+landUse”, enabling investigation of urban

land change effects). We recognize that high scenarios may not be the most likely scenarios. They are used here, because this research focuses on climate extremes, and high scenarios provide more data points on extreme cases, to support later trend identification and pattern comparison of influences from climatic and social systems and their interactions. Population exposures at the beginning of the 21st century (BOC; 1980-2005 summaries) (denoted as “historical”) were also analyzed as comparison baselines.”

Added a new SI Table 1 detailing the inner settings of all scenarios used in this research:

SI Table 1. Scenarios analyzed in this research, combining different climate, population, and land use conditions.

	Scenario Names	Climate Conditions	Population Conditions	Land Use Conditions
Key Scenarios	historical	BOC	BOC	BOC
	climate	EOC (RCP 8.5)	BOC	BOC
	climate+pop	EOC (RCP 8.5)	EOC (SSP 5)	BOC
	climate+pop+landUse	EOC (RCP 8.5)	EOC (SSP 5)	EOC (SSP 5)
Helper Scenarios	climate+landUse	EOC (RCP 8.5)	BOC	EOC (SSP 5)
	pop	BOC	EOC (SSP 5)	BOC

3. Table S1 - I’m a bit confused about the unit for the per capita exposure. ‘Days per person per year’ – why per person? This would mean that in an area of 1000 people and a per capita exposure of 30 the year has 30 days person-1 years-1*1000 persons = 30000 days per year ...

Edits: Not sure if I understand the question, but yes, “days per person per year” is the unit of the per capita annual exposure, because “per capita” means “per person”. Changed the figure title and caption to “SI Table 2. Comparison of annual population exposures to four climate extremes at BOC vs. EOC: Perspectives of both total and per capita exposures. For an example grid, if its annual per capita exposure is 30 days per person per year, and its population size is 1000, the grid’s annual total exposure is 30 * 1000 = 30,000 person days.”

4. Line 71 to 74 – how did you come to this conclusion?

Edits: The first two paragraphs of the Results section are a synthesis of all the later detailed results. Added “(details below)” at the end of the paragraph, now at lines 79-82.

5. Line 91 – is this not primarily an effect of ongoing urbanization, i.e, a higher percentage of people living in cities?

Edits: Yes, “more people will experience climate extremes as urban residents” (previously line 91) is because more people will live in urban areas.

6. Line 94 - mention that this is the case for all land uses/degrees of urbanization

Edits: Done.

7. Line 96 and following – if you do introduce denotations for your models now would be a good time to use them.

Edits: We took your recommendation to refer to the scenarios using reader friendly terms in place of technical model denotations as detailed above.

8. Fig 1 – I would suggest showing percentages instead of what you are doing now. (i.e., what is now 1 or -1 would become 100%). Might make things easier to read.

Edits: We chose ratio over percentage, because it’s important to use consistent metrics across the analyses of urban development densities, climate regions, and sizeable urban centers. For the latter two analyses, sometimes very large ratios emerge. It seemed easier to think about “A is 20 times B” rather than “A is 2000% B”.

9. Line 134 and following– how do these findings compare to other studies modelling exposure to extreme weather events?

Edits: As motivated in the Introduction (lines 44-49), a continental-scale spatial analysis is a novelty of this research.

10. Fig 2 Like with Fig 1 I suggest you use percentages.

Edits: Addressed above.

11. Fig 2 what models are you using to model climate effects and urban land use effects

Edits: Discussions of model choices are presented in the Methods section.

12. Line 181 please rephrase. ‘First, climate.’ Is not a sentence.

Edits: The full sentence here is “We identified two factors: First, climate. [...] Second, urban land patterns (...). [...]”

13. Fig 3 – units missing and y-axis descriptors missing in figure

Edits: The Y-axis is “urban land effects”, which is a ratio (i.e., dimensionless).

14. Fig 4 – text too small, unreadable.

Edits: This is an effect of having to fit the figure on one page and the image quality loss caused by converting to PDF. For the final submission, the original high-quality image will be submitted, and when (and if) published, readers can view it digitally with proper image quality and ability to zoom (especially, Nature Communications is an online only journal).

15. Line 290 missing ‘as’ before 2075

Edits: Done.

16. Fig S1 units missing (population per km²?)

Edits: Units were listed along with the title of each panel. “Urban land fraction” is dimensionless. The unit of “Population” is “thousand persons”.

17. Line 297 following. What’s the goodness of their models. While I understand that these models have been validated in previous studies, I feel some sort of discussion warranted. Also, what input parameters specifically are used from the SSP and RCP scenarios?

Edits: Now at lines 334-341: “The spatial urban land projections (SI Fig 1) were produced by a new data-science-based urban land modeling framework^{4,25,26} (namely, CLUBS-SELECT) which draws on 15 best available datasets on contemporary urbanization and its driving forces, and outperforms previous urban land models by reflecting (i) changes in urbanization styles over long time horizons, (ii) local-scale spatial variations in the urbanization process while maintaining global coverage (by modeling the world as 375 sub-national regions, each with one unique spatial model), and (iii) the emergence of new urban centers (in contrast to previous urban land models that generally expand existing urban centers for new development).”

18. Fig S2 – hard to compare as they are so similar. (Outside of the urban heat islands in the first row)

Edits: Figures highlighting our key findings and conclusions are the four figures in the main text. This SI figure is meant for (1) people who want to look at the raw spatial results as maps, and (2) reminding readers that at its core this research is a spatial analysis.

19. Line 310 where did you get data on a 5-year return period storm. Do BOC and EOC have different thresholds as the 5-year return period storm changes?

Edits: Clarified, now at lines 345-347: “a heavy rainfall is a precipitation event whose hourly precipitation amount is greater than or equal to the hourly precipitation amount of a 5-year return period storm at BOC calculated for every grid in the U.S. respectively”.

20. Fig S4 – legend (circle size) missing

Edits: Labeled example circle sizes for SI figures.

21. Around Line 352 and following – the figure S4 only shows 3 scenarios, not the five you discuss here

Edits: Corrected the typo – we have not five but four key scenarios, as shown in the scenario table presented above. The figure shows three out of the four, leaving out the climate+pop scenario, as this research argues, to properly consider urbanization, both population and land use changes should be considered (i.e., climate+pop+landUse).

Reviewer #2:

Dear Drs. Gao and Bukovsky,

The paper entitled, “Urban land patterns can moderate population experiences of 21st-century climate change,” explores the effect of urbanization and climate change on population exposure to hot days, cold days, rainfall, and severe storms in urbanized areas of the United States.

Overall, I find this work to be important, timely, and relevant to the journal. I thank the authors for a well structured manuscript with methods that are easy to follow.

I have no major comments at this time. The methods are sound and the results are appropriately interpreted. There is some clarification needed, as explained below, but this is very minor.

Thank you for the affirmative evaluation of our work.

The Introduction, specifically focusing on literature review, is very short and seems to be missing information.

For example, on lines 26-27, you mention that urbanization increases population exposure. It would be extremely helpful to discuss why this is the case as you then discuss modifying land use in further parts of the introduction. What about urban land use specifically causes these patterns of exposure?

Edits: Added new texts to the Introduction section, to expand discussions of underlying mechanisms.

At lines 24-30: “Over the coming decades, climate change is projected to increase the frequency and the intensity of extreme weather², and global urbanization is expected to bring about further population concentration and urban land expansion^{3,4}. These mechanisms interact, and as a result, urbanization has usually been reported to escalate population exposures (frequency and intensity) to climate extremes⁵⁻⁷ (e.g., when urban lands enhance climate extremes at where vulnerable population concentrates); however, urban land patterns are human designs that can be changed.”

At lines 50-56: “Existing literature on future population’s vulnerability to climate extremes has focused on temperature^{7,15,16}. However, patterns of extreme precipitation and convective environments will also evolve, as both climate change and urbanization modify

future land-atmosphere dynamics¹⁷⁻²². Effects of urban heat island (UHI) not only strengthen sensible heat fluxes to increase temperature, but also enhance surface roughness and atmospheric circulations to influence the frequency and the intensity of severe convective storms²³, including extreme precipitation and thunderstorms.”

Lines 141-142: While I agree that some areas will increase in vulnerability because of lack of acclimatization, past experiences, and protective measures, do you consider the southeast in this position? Much of the SE US has dealt with heat stress, especially when including humidity, every summer.

Edits: Rewrote the paragraph for clarity, now at lines 155-166: “For hot days, the ranking of regions by their exposure counts shuffle from BOC to EOC: Some regions that do not rank highly at BOC are anticipated to become new hotspots of heat exposure at EOC (Fig 2, SI Fig 4), e.g., the Southeast is expected to substantially surpass the Southwest to become the most heat-stressed region in the country, the Southern Great Plains catch up to the Southwest, while the Northeast and the Midwest also experience considerable increases in heat exposure. The drastic changes in the ranking of regions are important, since planning and distributing resources according to past priorities might lead to additional vulnerabilities for the future, especially for the regions rising in priorities.”

Line 161: What societal factors would specifically cause less exposure?

Edits: Rephrased the sentence, now at lines 183-185: “the Northwest and the Northern Great Plains are anticipated to have the least population exposures to climate extremes due to both climatic and urban processes”.

Elaborated about potential causes of reduced exposure where the finding is first presented, at lines 110-122: “Surprisingly, urban land expansion can also decrease average exposures to climate extremes in some cases. For example (SI Table 2), urban land expansion in mid-to-high-density developments decreases the EOC per capita exposure to heavy rainfalls. This change in exposure might be caused by physical land-atmosphere mechanisms (e.g., UHI effects can strengthen convective inhibition over densely developed urban areas⁸, which can decrease the frequency of extreme precipitation events to counteract the broadly-occurring enhancing effect of climate change), and/or spatial population-urban-

land dynamics (e.g., many densely developed urban areas are not residential and hence show low population exposures). Identifying what patterns and mechanisms can reduce population exposures to climate extremes, and using that knowledge to inform urban designs, are our planned next research steps. The significance of this work is that the results show carefully designed urban land patterns are able to practically moderate instead of intensify the nation's future experiences of climate extremes (more evidence in later sections).”

Lines 175-177: You mention later in the manuscript that this is an area for future research, but is there anything you can generally suggest for how urban land can be used to moderate climate extremes?

Edits: Edits addressing the above comment also clarify for this one.

Elaborated about potential causes/directions for future work when discussing city-level reduced exposure to hot days, at lines 230-242: “For temperature extremes, urban land expansion generally leads to warming, due to UHI effects, but exceptions exist for a few urban centers, to reduce exposure to hot days and increase exposure to cold days. Though these exceptions are of small magnitudes, their existence showcases the possibility of using urban land effects to moderate, or at least not horribly amplify, climate effects. Especially for hot days, a total of seven urban centers’ land changes reduce exposures, i.e., Savannah, GA, Los Angeles, CA, Boise, ID, Seattle, WA, San Francisco, CA, Salt Lake City, UT, and Virginia Beach, VA, with their urban land effects on population exposures to hot days ranging from 0.92 to 0.48. This is NOT to say that the current urban land patterns of these cities already moderate population exposures, but rather their urban land changes simulated in our hypothesized high urbanization scenario show that effect. Investigations of what mechanisms drive the exposure reductions are planned for our next research. We will consider both physical factors (e.g., changes in evapotranspiration, albedo, sea breeze) and population-land dynamics (e.g., infill development, migration).”

Expanded texts on future research plans in the Discussion section, at lines 288-292: “It is therefore worthwhile and beneficial to identify urban land patterns and related physical and socioeconomic mechanisms that can reduce population exposures to climate extremes over the long term, and use such understandings to guide urban designs for the future.”

Reviewer #3:

General comments: This is a paper that will likely be of great interest to readers. The objective is to examine how changes in urban land, population, and climate will influence spatial patterns of population exposure to climate stressors (cold and hot temp extremes, heavy precip, and severe thunderstorm events) in the future. Results diverge from what's commonly perceived in the field in that urban land can in some cases reduce population exposure to climate extremes, including hot temp extremes. Further, findings indicate that spatial patterns rather than total quantity of urban land played a larger role. One potential limitation is the validity of the effect ratio approach. While I'm struck by the simple elegance of this measure, the authors don't currently reference others in the field who have used this approach to tease out land use from climatic changes. However, it appears that the effect ratio is only measuring the relative contribution of climate change or urban land patterns on population exposure. Why not take the study a step further and incorporate an absolute measure that shows the risk difference (relative and absolute measure) that shows the actual number for population exposed to quantify the excess contribution of climate change or urban land on population exposure to climate extremes. The methods section could benefit from further sharpening. There is a great deal of study required for the paper involving keeping track of the various scenarios, spatial aggregation scales, and climate models. More fine-tuning to enhance readability in the naming conventions of the various scenarios would be helpful.

Thank you for helping us improve this work. We added new texts justifying and explaining the effect ratio approach, and simplified references to the various scenarios (details below).

Recommendation: Major revisions

Introduction

It's not clear that there has been a strong climate signal implicated in severe thunderstorms and wondered why the authors decided to include this in the line-up of extremes rather than say drought events as a compliment to heavy precip? Is it more of a conceptual control that removes climate changes and just looks at the influence of population and land use changes (only)?

Edits: Conceptual control was certainly a consideration, and also scale – we wanted a diverse line-up of disasters that respond to urban land changes, since the end point is population exposure.

Added more rationale, at lines 50-57: “Existing literature on future population’s vulnerability to climate extremes has focused on temperature^{7,15,16}. However, patterns of extreme precipitation and convective environments will also evolve, as both climate change and urbanization modify future land-atmosphere dynamics¹⁷⁻²². Effects of urban heat island (UHI) not only strengthen sensible heat fluxes to increase temperature, but also enhance surface roughness and atmospheric circulations to influence the frequency and the intensity of severe convective storms²³, including extreme precipitation and thunderstorms. Here we examine hot days, cold days, heavy rainfalls, and STEs.”

A text box or table mapping the different scenarios used might provide a helpful quick reference for the reader

Edits: Changed the notations of the scenarios throughout the manuscript to be more reader friendly, including texts, figures, tables, SI, etc. For example, at lines 58-72: “We analyzed three future scenarios combining different climate, land use, and population conditions at the end of the 21st century (EOC; 2075-2100 summaries) for spatial population exposures to the four climate extremes at a 25-km resolution (more information in Methods and SI Table 1): (i) high greenhouse gas warming with no population nor urban land change (denoted as “climate”, showing effects from climate change only), (ii) high greenhouse gas warming and high population growth with no urban land change (“climate+pop”, the current common practice), and (iii) high greenhouse gas warming, high population growth, and high urban land expansion (“climate+pop+landUse”, enabling investigation of urban land change effects). We recognize that high scenarios may not be the most likely scenarios. They are used here, because this research focuses on climate extremes, and high scenarios provide more data points on extreme cases, to support later trend identification and pattern comparison of influences from climatic and social systems and their interactions. Population exposures at the beginning of the 21st century (BOC; 1980-2005 summaries) (denoted as “historical”) were also analyzed as comparison baselines.”

Added a new SI Table 1 detailing the inner settings of all scenarios used in this research:

SI Table 1. Scenarios analyzed in this research, combining different climate, population, and land use conditions.

	Scenario Names	Climate Conditions	Population Conditions	Land Use Conditions
Key Scenarios	historical	BOC	BOC	BOC
	climate	EOC (RCP 8.5)	BOC	BOC
	climate+pop	EOC (RCP 8.5)	EOC (SSP 5)	BOC
	climate+pop+landUse	EOC (RCP 8.5)	EOC (SSP 5)	EOC (SSP 5)
Helper Scenarios	climate+landUse	EOC (RCP 8.5)	BOC	EOC (SSP 5)
	pop	BOC	EOC (SSP 5)	BOC

Results

Should there be a table or figure referenced in the first couple of paragraphs of the Results section?

Edits: The first 2-3 paragraphs of the Results section are setting up the stage for the later breakdown of the results. The few numbers mentioned here are all total numbers, and there is no further detail to illustrate or list.

The authors state: “We capture effects of changes in climate, urban land, and population over the 21st century as ratios between exposure counts (ECs) from different scenarios, because such ratios can robustly separate effects of climate, urban land, and population (more explanations in Methods).” More rational as to how and why these ratio measures adequately measure the effects of climate change from BOC to EOC is needed

Edits: Added new texts explicitly reasoning the use of effect ratios, and added one numerical example, at lines 417-451: “To isolate effects on population exposure to climate extremes from the three variables respectively (climate, land use, population), we compare the stratified aggregate exposure counts (ECs) from the four above mentioned scenarios with each other (SI Fig 4). In theory, because exposures are defined as the product of population size and average frequency of each climate extreme, ratio rather than subtraction based metrics would more naturally separate effects from climate, land use, and population changes, while in practice, subtraction based metrics are broadly used. Here we tested both kinds of effect metrics. In this research, multiple different pairs of scenarios may be used to measure effects of changes in the same variable. For example, for

measuring land use change effects, the scenario pairs (climate+pop+landUse, climate+pop) and (climate+landUse, climate) are both conceptually appropriate. However, when using subtraction based metrics, $(EC_{\text{climate+pop+landUse}} - EC_{\text{climate+pop}})$ and $(EC_{\text{climate+landUse}} - EC_{\text{climate}})$ result in largely different values, e.g., for the national total exposure to hot days, $(EC_{\text{climate+pop+landUse}} - EC_{\text{climate+pop}})$ is 5.9 billion person days, and $(EC_{\text{climate+landUse}} - EC_{\text{climate}})$ is 2.4 billion person days; when using ratio based metrics, $(EC_{\text{climate+pop+landUse}} / EC_{\text{climate+pop}})$ and $(EC_{\text{climate+landUse}} / EC_{\text{climate}})$ result in nearly identical values, e.g., also for the national total exposure to hot days, $(EC_{\text{climate+pop+landUse}} / EC_{\text{climate+pop}})$ is 1.28, and $(EC_{\text{climate+landUse}} / EC_{\text{climate}})$ is 1.27. This finding is consistent with the theoretical reasoning that for multiplication-based metrics (e.g., exposure counts), ratio based metrics can clearly separate effects from different amplifying factors. Essentially, the fact that $(EC_{\text{climate+pop+landUse}} - EC_{\text{climate+pop}})$ is much larger than $(EC_{\text{climate+landUse}} - EC_{\text{climate}})$ is because the former captured some population effects while aiming to capture only urban land effects. To confirm the robustness of ratio based effect measures broadly holds, we generated two helper scenarios: “climate+landUse” (capturing effects of changes in both climate and land use, but assuming no population change), and “pop” (combining historical climate and land use with future population conditions), and used all possible scenario pairs for calculating climate, land use, and population effects across all national totals, urban development densities, climate regions, and sizeable urban centers. The results show, empirically, the robustness of the effect ratios always holds. Therefore, we finalize our definitions of effects using only the four key scenarios, and define climate effects as $(EC_{\text{climate}} / EC_{\text{historical}})$, land use effects as $(EC_{\text{climate+pop+landUse}} / EC_{\text{climate+pop}})$, and population effects as $(EC_{\text{climate+pop}} / EC_{\text{climate}})$.”

SI Table 1. Can the authors better orient the reader for the inclusion of Total Exposure and Per Capita Exposure and how they arrive at billion person days vs days per person per year? Is the table meant to reflect total change in population exposure between the two exposure scenarios and across climate extremes??? A range would be more helpful in reporting (e.g., hot days is not developed areas by end of century 26% (18, 32)

Edits: Changed the figure title and caption to “SI Table 2. Comparison of annual population exposures to four climate extremes at BOC vs. EOC: Perspectives of both total and per capita exposures. For an example grid, if its annual per capita exposure is 30 days per person per year, and its population size is 1000, the grid’s annual total exposure is $30 * 1000 = 30,000$ person days.”

Re: stationarity in population change estimates lending to insufficiencies in analyzing potential changes in spatial patterns - pg 6 line 116?

Edits: Happy to make any edit you suggest but not sure we understand the question here.

Fig 2 -can the authors comment on the large ratios for the climate effects and hot days 27.56, 21.21

Edits: The Northern Great Plains and the Northeast conventionally are not particularly hot regions. If an area used to have 1 hot day per year at BOC, a climate effect of 25 means the area is anticipated to have 25 hot days per year at EOC.

Fig 3 conveys a large amount of information, but takes a lot of studying for the reader to be able to derive meaning. Wonder if there might be a more effective way to visualize these results.

Edits: Thank you for the comment. There is indeed a lot of info to show, and we spent a lot of time and consideration on designing this figure. Happy to make any edit you suggest.

Especially, the 214 urban land projections we used in this research are produced by a novel data-driven model^{4,24,25} 215 (more information in Methods). [incomplete?]

Edits: Happy to make any edit you suggest but not sure we understand the question here.

Changed the punctuation for the paragraph, please see if it’s clearer, now at lines 246-257:

“These results indicate a loose association between urban land effects and climate trends, suggesting the potential of urban land patterns to moderate population exposures to climate extremes at the city level is not dependent on climate effects and is non-trivial.

Especially, the urban land projections we used in this research are produced by a novel data-driven model^{4,25,26} (more information in Methods): While data-driven models are good at capturing various underlying processes and their transitions from data, in this research the projected future urban land patterns would not have reflected conscious land use

designs for long-term climate resilience, because such designs have rarely happened in recent past decades and hence have little reflection in existing data. In the future, as we more intentionally design our urban and regional land use patterns, their desirable moderating effects on population exposures to climate extremes will likely be larger than what have been observed in this research.”

Fig 4 - nice and effective visual

Thanks!

Discussion

Lacks a strengths/limitations section

Edits: Add a paragraph on limitations, at lines 459-467: “While using state-of-the-art methods, this work is still affected by some methodological limitations, primarily: First, the climate simulations used only one model, configuration, and forcing combination. Though a well-documented, well-regarded, and widely-used choice, further research examining the effects of model structural uncertainties on studying society-climate interactions is important. A detailed discussion about potential future research in this direction can be found in our recent paper⁸. Second, due to the insufficiencies of current long-term spatial population models and projections discussed earlier, new model development is needed, before in-depth examinations of how spatial population patterns interact with urban land and climate can be possible. We are currently working on these research directions.”

Methods

Is there an inherent benefit/advantage of using the Weather Research and Forecasting (WRF) model?

Edits: Aside from it being a well-documented, well-regarded, and widely-used model, there is no inherent benefit/advantage. Also, this work leveraged existing climate simulations from a published study which was done using WRF with an already vetted configuration and boundary condition dataset, to save computational costs.

There are a number of absolute and percentile-based thresholds for defining extreme temperature events (e.g., above the 95th or below the 5th percentile based on the climatology of a given

geographic area). The justification for the cold and hot temp days is not well substantiated by the literature and why not include a couple of additional metrics in a sensitivity analysis?

Edits: We wanted to examine a range of climate extremes that may affect future urban lives. Temperature extremes, out of the four we examined, are the most studied and well covered by existing literature. The commonly-used, simple threshold, >35°C, generated reasonable results that are consistent with previous reports while adding new dimensions of information; hence, it didn't seem necessary to analyze definition sensitivity. However, when analyzing heavy rainfalls, we also started with a commonly-used, simple threshold, >1 inch of precipitation per hour, but found that disregarding baseline precipitation levels at different locations tremendously biased the results, so we switched to the computationally-expensive frequency-based definition of heavy rainfalls calculated for every grid in the U.S. respectively.

Re: different aggregation strata for exposure maps. How did the authors arrive at the 20-KM - 100-km buffers for cities with a population larger than 300k? There still seems to be spatial overlap among large cities in the Northeast and out west in California.

Edits: Added explanations at lines 395-401: “Different buffer radiuses are applied to different city population size categories, as cities with larger population sizes occupy more land areas and interact with larger regions. Using a uniform radius would either cut off parts of large cities, or dilute effects of small cities with too much regional contexts of little land change. The specific radius for each city population size category was chosen so that urban centers in the category will be completely encompassed by their buffers at EOC, i.e., after the maximum urban land expansion has taken place.” “Resolving merging boundaries of the buffers, we have 109 urban centers across the continental U.S. (SI Fig 3).”

Re: scenarios - for the different scenarios - can the authors come up with a better naming convention to aid interpretability for the reader, there are a lot of acronyms to keep track with that are not intuitive (RCP8.5_SSP5_SSP5, RCP8.5_hist_SSP5, RCP8.5_SSP5_hist, RCP8.5_hist_hist) and then spell out these scenarios in figures/tables + captions?

Edits: Changed the scenario denotations as addressed above.

Re: effect ratios - it's not clear if this is a standard approach already used in the field to compare exposure count climate scenarios.

Re: To confirm the robustness of measuring effects as ratios, we added a dummy scenario 362 hist_hist_SSP5 to our scenario pool, and used all possible scenario pairs for calculating climate, 363 land use, and population effects as ratios across all national totals, urban development densities, 364 climate regions, and sizeable urban centers. The robustness of the effect ratios always holds. Can the authors point the reader to these results in supplemental materials? There are no citations providing justification of this approach.

Edits: As noted earlier, added new texts explicitly reasoning the use of effect ratios, and added one numerical example, at lines 417-451.

REVIEWERS' COMMENTS

Reviewer #1 (Remarks to the Author):

Review of the revised manuscript

"Urban Land Patterns Can Moderate Population Exposures to 21st-Century Climate Extremes"

Manuscript ID: NCOMMS-22-35095A.

General comments:

The Authors did a wonderful job revising the manuscript and answering my concerns. I am now happy to recommend publications outside of a few minor and one medium issues.

1) I am still confused about the per capita metric. Are you sure you mean exposure per capita and not exposer of the average person in the US?

If I read this correctly you simply divide your exposure metric (that has a unit of bil person days following figure 2) by the population count. This gives you the number of exposure days each person experiences on average, but NOT exposure per person. I assume this is just a language error. Per person implies 1/person as is used e.g., in per capita GDP. Please adjust the language accordingly

2) Line 29 (in the document with tracked changes) "e.g., when urban lands enhance climate extremes at where vulnerable population concentrates" delete the "at"

3) Line 51: I would suggest adding a primarily. "Existing literature on future population's vulnerability to climate extremes has primarily focused on temperature"

4) Line 67: consider letting the reader know that the high scenario refers to RCP 8.5 and SSP 5

5) Line 89: When quantifying the impact of land use, you use EC_Climate+Pop as a baseline, for population change you use EC_Climate. Is there a specific reason why you aren't using EC_Climate+LandUse? Is such a scenario not possible?

6) Line 205 "But what are?" is rather dramatic language. I would suggest rephrasing. I'm also not sure if it should be "what is?" if you want to keep that sentence.

7) Line 206 and 211: I would suggest changing "First, Climate." And "Second, urban land patterns []." To "First, Climate:" And "Second, urban land patterns []:"

8) Line 206 should be "Not only do some low level correlations exist ..."

9) Line 234 and the following: Awesome! I really like this addition. Thank you.

Reviewer #3 (Remarks to the Author):

The revised manuscript is responsive to each reviewer critique and much improved. I have no additional comments or suggestions.

Dear reviewers,

Thank you for taking the time to help us improve this manuscript! Below we address your comments one by one. Hope you find the changes satisfactory.

Best regards,

Jing Gao & Melissa Bukovsky

Reviewer #1:

General comments:

The Authors did a wonderful job revising the manuscript and answering my concerns. I am now happy to recommend publications outside of a few minor and one medium issues.

Thank you for helping us improve this work.

1) I am still confused about the per capita metric. Are you sure you mean exposure per capita and not exposer of the average person in the US?

If I read this correctly you simply divide your exposure metric (that has a unit of bil person days following figure 2) by the population count. This gives you the number of exposure days each person experiences on average, but NOT exposure per person. I assume this is just a language error. Per person implies 1/person as is used e.g., in per capita GDP. Please adjust the language accordingly

Edits: Your reading of the “per capita exposure” definition is correct. We used the word “per capita” in the same way as “per capita GDP”. According to the World Bank, “GDP per capita is the sum of gross value added by all resident producers in the economy plus any product taxes (less subsidies) not included in the valuation of output, divided by mid-year population.” Per capita GDP = total GDP/population, and per capita exposure = total exposure/population.

2) Line 29 (in the document with tracked changes) “e.g., when urban lands enhance climate extremes at where vulnerable population concentrates” delete the “at”

Edits: Changed to “e.g., when urban lands enhance climate extremes at places where vulnerable population concentrates”.

3) Line 51: I would suggest adding a primarily. “Existing literature on future population’s vulnerability to climate extremes has primarily focused on temperature”

Edits: Done.

4) Line 67: consider letting the reader know that the high scenario refers to RCP 8.5 and SSP 5

Edits: Changed to “We recognize that high scenarios (details in Methods) may not be the most likely scenarios.” During the last round, multiple reviewers suggested we introduce RCP8.5 and SSP5 only in Methods, and use combinations of “climate”, “pop”, and “landUse” to denote our scenarios. This tremendously improved the readability of the manuscript, so we did not add RCP8.5 and SSP5 back to the Introduction, but rather a reference to Methods.

5) Line 89: When quantifying the impact of land use, you use EC_Climate+Pop as a baseline, for population change you use EC_Climate. Is there a specific reason why you aren’t using EC_Climate+LandUse? Is such a scenario not possible?

SI Table 1. Scenarios analyzed in this research, combining different climate, population, and land use conditions at the beginning (BOC) and the end (EOC) of the 21st century, using the Shared Socioeconomic Pathways and the Representative Concentration Pathways (the SSP-RCP scenario framework).

	Scenario Names	Climate Conditions	Population Conditions	Land Use Conditions
Key Scenarios	historical	BOC	BOC	BOC
	climate	EOC (RCP 8.5)	BOC	BOC
	climate+pop	EOC (RCP 8.5)	EOC (SSP 5)	BOC
	climate+pop+landUse	EOC (RCP 8.5)	EOC (SSP 5)	EOC (SSP 5)
Helper Scenarios	climate+landUse	EOC (RCP 8.5)	BOC	EOC (SSP 5)
	pop	BOC	EOC (SSP 5)	BOC

Edits: Analytically, the scenario climate+landUse is possible, and was produced as a helper scenario in our analysis (see SI Table 1 above) for evaluating the robustness of ratio based effect measures (complete details at lines 406-435). It was not used in the final definitions,

because (1) the ratio based effect measures are robust across scenario combinations, that is, $(EC_{\text{climate+pop+landUse}} / EC_{\text{climate+pop}})$ and $(EC_{\text{climate+landUse}} / EC_{\text{climate}})$ are nearly identically valued, (2) since climate+pop is the current status quo, climate+landUse is not commonly seen in existing literature, and (3) climate+pop+landUse rather than climate+landUse is the optimal practice to recommend.

6) Line 205 “But what are?” is rather dramatic language. I would suggest rephrasing. I’m also not sure if it should be “what is?” if you want to keep that sentence.

Edits: This part of Results presents the most noteworthy findings. The sentence is immediately followed by the statement “we identified TWO factors”, hence the plural form is used. Now at lines 196-198: “Moreover, urban land effects show no correlation with urban land total amounts nor change amounts (Fig 3), suggesting neither is a key determinant of urban land effects at the city level, but what are influential? We identified two factors:”

7) Line 206 and 211: I would suggest changing “First, Climate.” And “Second, urban land patterns [].” To “First, Climate:” And “Second, urban land patterns []:”

Edits: The overall structure here is “We identified two factors: First, climate. [...] Second, urban land patterns. [...]”. Using colons after naming each of the two factors would create a nested colon-based structure.

8) Line 206 should be “Not only do some low level correlations exist ...”

Edits: Only one type of correlation is present, hence the singular rather than plural form seems more warranted.

9) Line 234 and the following: Awesome! I really like this addition. Thank you.

Thank you!

Reviewer #3:

The revised manuscript is responsive to each reviewer critique and much improved. I have no additional comments or suggestions.

Thank you for helping us improve this work.